# Hydrophobically Modified Isosorbide Dimethacrylates as a Bisphenol-A (BPA)-Free Dental Filling Material

**DOI:** 10.3390/ma14092139

**Published:** 2021-04-22

**Authors:** Bilal Marie, Raymond Clark, Tim Gillece, Seher Ozkan, Michael Jaffe, Nuggehalli M. Ravindra

**Affiliations:** 1Interdisciplinary Program in Materials Science and Engineering, New Jersey Institute of Technology, Newark, NJ 07012, USA; bm253@njit.edu (B.M.); jaffe@njit.edu (M.J.); 2Ashland Specialty Ingredients, Bridgewater, NJ 08807, USA; rbclark@ashland.com (R.C.); Tgillece@ashland.com (T.G.); Sozkan@ashland.com (S.O.)

**Keywords:** isosorbide dimethacrylates, dental filling material, hydrophobic, bio-based, bisphenol-A (BPA)

## Abstract

A series of bio-based hydrophobically modified isosorbide dimethacrylates, with *para-*, *meta-*, and *ortho-* benzoate aromatic spacers (ISBGBMA), are synthesized, characterized, and evaluated as potential dental restorative resins. The new monomers, isosorbide 2,5-bis(4-glyceryloxybenzoate) dimethacrylate (ISB4GBMA), isosorbide 2,5-bis(3-glyceryloxybenzoate) dimethacrylate (ISB3GBMA), and isosorbide 2,5-bis(2-glyceryloxybenzoate) dimethacrylate (ISB2GBMA), are mixed with triethylene glycol dimethacrylate (TEGDMA) and photopolymerized. The resulting polymers are evaluated for the degree of monomeric conversion, polymerization shrinkage, water sorption, glass transition temperature, and flexural strength. Isosorbide glycerolate dimethacrylate (ISDGMA) is synthesized, and Bisphenol A glycerolate dimethacrylate (BisGMA) is prepared, and both are evaluated as a reference. Poly(ISBGBMA/TEGDMA) series shows lower water sorption (39–44 µg/mm^3^) over Poly(ISDGMA/TEGDMA) (73 µg/mm^3^) but higher than Poly(BisGMA/TEGDMA) (26 µg/mm^3^). Flexural strength is higher for Poly(ISBGBMA/TEGDMA) series (37–45 MPa) over Poly(ISDGMA/TEGDMA) (10 MPa) and less than Poly(BisGMA/TEGDMA) (53 MPa) after immersion in phosphate-buffered saline (DPBS) for 24 h. Poly(ISB2GBMA/TEGDMA) has the highest glass transition temperature at 85 °C, and its monomeric mixture has the lowest viscosity at 0.62 Pa·s, among the (ISBGBMA/TEGDMA) polymers and monomer mixtures. Collectively, this data suggests that the ortho ISBGBMA monomer is a potential bio-based, BPA-free replacement for BisGMA, and could be the focus for future study.

## 1. Introduction

Dental amalgam and resin-based composites are commonly used dental filling materials. The recent US-FDA epidemiological review of exposure to dental amalgam mercury did not find sufficient evidence correlating adverse health outcomes with exposure to dental amalgam mercury [1]. However, the negative perception of amalgam, lower aesthetic appeal [2], and the Minamata convention to minimize mercury production and impact [3], are leading reasons that resin-based composites are preferred.

Dental resin composites consist of a polymeric resin, filler particles, and a coupling agent. The polymeric resin is largely made up of hydrophobic dimethacrylates that crosslink into a three-dimensional network. These polymeric networks should have a high degree of monomeric conversion, low water uptake, low polymerization shrinkage, and good mechanical properties [4,5].

The most common dental restorative dimethacrylate monomer is 2,2-bis [4-(2-hydroxy-3-methacryloyloxypropoxy)phenyl] propane (BisGMA). Its structure imparts a high polymeric modulus, low shrinkage, and strong adhesion to the tooth enamel [6,7,8]. It is highly viscous and is typically mixed with the dental diluent, triethylene glycol dimethacrylate (TEGDMA). However, TEGDMA is reported to increase water sorption and polymerization shrinkage [9,10]. In contrast, 1,6-bis(2-methacryloxy-2-ethoxycarbonylamino)-2,4,4-trimethylhexane (UDMA) is a urethane dimethacrylate with lower viscosity in comparison to BisGMA, and exhibits good durability and adhesion to the tooth enamel [11,12].

Dimethacrylates exhibit excellent properties in resin dental restoration. However, dental dimethacrylate monomers, and the resulting degradation products, were demonstrated to exhibit a cytotoxic and genotoxic effect [13,14]. Further, prior investigations suggest that TEGDMA and UDMA can induce apoptosis in dental pulp [15,16]. Trace amounts of Bisphenol A (BPA), a suspected estrogen mimic, can be present in BisGMA and elute into the oral environment [17,18]. Increased levels of BPA exposure have been linked to various prenatal, childhood, and adult adverse health outcomes. These include, but are not limited to, reduced fertility, altered childhood behavior and neurodevelopment, type-2-diabetes, cardiovascular disorders, inflammation, and altered gene expression [19]. Hence, there has been a growing interest in developing safer alternative materials.

Isosorbide is a sugar-based molecule derived from starch and is classified as “generally recognized as safe”. It is made up of two *cis*-fused tetrahydrofuran rings, which are nearly planar, and contains two hydroxyls at positions 2 and 5, as shown in Figure 1 [20,21]. Isosorbide has found potential applications in the biomedical field, including aspirin prodrugs [22,23], substrates for human butyrylcholinesterase [24], tissue engineering scaffolds [25,26], and dental restorative materials [27,28,29,30].

Some of the studies on isosorbide for dental materials include the synthesis and evaluation of the BisGMA analogue, 1,4:3,6-dianhydro-d-sorbitol 2,5-bis(2-hydroxy-3-methacylolxypropoxy) (ISDGMA) as reported by Łukaszczyk et al., and Shin and coworkers [27,28]. Duarte et al., have studied the synthesis of an isosorbide urethane dimethacrylate (IS-UDMA) [29]. Vasifihasel et al. and Łukaszczyk and co-workers developed ethoxylated isosorbide dimethacrylates (ISETDMA) as a dental diluent [30,31]. Jun et al. developed isosorbide dimethacrylates based on isocyanoethyl methacrylates and evaluated their performance as dental sealants [32]. Kim et al. reported on isosorbide 2,5-bis(propoxy) dimethacrylate (ISOPMA) and evaluated it as a dental resin composite in comparison to ISDGMA [33].

The aim of this study is to synthesize and investigate the development of new hydrophobically-modified isosorbide dimethacrylate monomers, using *para*, *meta*, and *ortho*-substituted hydroxy benzoates as hydrophobic spacers. The new monomers are 1,4:3,6-dianhydro-d-sorbitol 2,5-bis [4-(2-hydroxy-3-methacryloyloxypropoxy)benzoate] (ISB4GBMA), 1,4:3,6-dianhydro-d-sorbitol 2,5-bis [3-(2-hydroxy-3-methacryloyloxypropoxy)benzoate] (ISB3GBMA), and 1,4:3,6-dianhydro-d-sorbitol 2,5-bis[2-(2-hydroxy-3-methacryloyloxypropoxy)benzoate] (ISB2GBMA). Their chemical structures are shown in Figure 2. The biobased carbon content of these monomers is about 18%.

This new series of isosorbide dimethacrylates is evaluated as copolymers with TEGDMA, and the performance attributes as dental restorative resins are compared to copolymers of ISDGMA/TEGDMA and BisGMA/TEGDMA. The chemical structures of ISDGMA, BisGMA, and TEGDMA are depicted in Figure 3.

This investigation was performed under the hypothesis that hydrophobic ISBGBMA (bio-based hydrophobically modified isosorbide dimethacrylates) monomers, in contrast to hydrophilic ISDGMA, will result in lower water uptake and improved mechanical properties, when co-polymerized with TEGDMA for dental applications.

This concept was previously addressed by the Jaffe group [34]. In that study, the water uptake of an isosorbide thermoset was reduced, and its mechanical properties were enhanced when a hydrophobic moiety was incorporated into the backbone of the polymer. We also theorize the ISBGBMA monomers to have properties similar to BisGMA. The findings of this investigation will provide value for further isosorbide dimethacrylate development.

## 2. Materials and Methods

Isosorbide (98%), methyl 4-hydroxybenzoate (≥99%), methyl 3-hydroxybenzoate (99%), methyl 2-hydroxybenzoate (≥99%), methacrylic acid (MAA, 99%), bisphenol A glycerolate dimethacrylate (BisGMA, >98%), triethylene glycol dimethacrylate (TEGDMA, 95%), camphorquinone (CQ, 97%), 2-(dimethylamino)ethyl methacrylate (DMAEMA, 98%), and triphenylphosphine (TPP, 99%) were acquired from Sigma-Aldrich (St. Louis, MO, USA) and used without further purification. Allyl bromide (99%), 3-chloroperbenzoic acid (mCPBA, 75%), ethylcarbodiimide hydrochloride (EDC, 99%), 4-(dimethylamino)pyridine (DMAP, 98%), 4-methoxyphenol (MeHQ), phenothiazine (98%), *N*,*N*-dimethylformamide (DMF, 99%), methanol (MeOH, 99%), methylene chloride (DCM, 99%), anhydrous potassium carbonate (K_2_CO_3_, 99%), and sodium hydroxide (NaOH, 98%) were obtained from Oakwood Chemical (Estill, SC, USA) and used as received. Gibco DPBS (1X) pH 7.1 was obtained from Thermo Fisher Scientific (Waltham, MA, USA) and used as supplied.

Fourier Transform Nuclear Magnetic Resonance spectroscopy data were obtained using an Agilent 400 MHz FT-NMR spectrometer (Santa Clara, CA, USA). Attenuated Total Reflectance Fourier-Transform infrared (ATR-FTIR) spectroscopy was performed on a Thermo Fisher Nicolet iS10 instrument (Waltham, MA, USA). Contact angle measurements with water were carried out using a Kruss—DSA30S drop shape analyzer (Hamburg, Germany). Viscosity measurements were performed using a Brookfield AMETEK DV2D LV viscometer at 200 RPM (Middleborough, MA, USA). Glass transition temperatures (T_g_) were measured with a TA Instruments Q800 DMA (New Castle, DE, USA). Flexural strength was measured with a TA.XT Plus texture analyzer (Hamilton, MA, USA). The logarithm of the 1-octanol/water partition coefficient was estimated using ChemDraw Prime 17.1 (Cambridge Software from Perkin Elmer). One-way ANOVA with a Tukey post-hoc test was used for statistical analysis, with a significance level of *p* < 0.05 using IBM SPSS Statistics software, version 27.

### 2.1. Synthesis

Since the synthesis of the para-, ortho-, and meta-substituted isosorbide 2,5-bis(glyceryloxybenzoate) dimethacrylates is similar, the para-substituted isosorbide 2,5-bis(4-glyceryloxybenzoate) dimethacrylate reaction scheme is given as an example in Figure 4.

Starting with methyl 4-hydroxybenzoate, allyl bromide is added in the presence of anhydrous potassium carbonate and DMF. The resulting product is distilled to produce methyl 4-allyloxybenzoate. After saponification with sodium hydroxide in methanol, 4-allyloxybenzoic acid is esterified with isosorbide in the presence of EDC and DMAP in DCM to make isosorbide 2,5-bis(4-allyloxybenzoate).

Isosorbide 2,5-bis(4-allyloxybenzoate) is further reacted with MCPBA to generate isosorbide 2,5-bis(4-glycidyloxybenzoate) (60% yield, >98% pure, mp 112–113 °C). ^1^H NMR (CDCl_3_-δ, ppm): 2.78 (2H, -CH oxirane), 2.93 (2H, -CH oxirane), 3.37 (2H, -CH oxirane), 3.9–4.1 (6H, 4H -CH_2_ isosorbide, 2H-CH_2_ glycidyloxy), 4.29–4.31 (2H, -CH_2_ glycidyloxy), 4.67 (1H, -CH isosorbide), 5.03 (1H, -CH isosorbide), 5.39 (1H,-CH isosorbide), 5.46 (1H-CH isosorbide), 6.94 (4H, -CH aromatic), 7.9 (2H, -CH aromatic), and 8.0 (2H, -CH aromatic).

The final dimethacrylate monomer (ISB4GBMA) is attained upon reacting 20 g (40 mmol) of isosorbide 2,5-bis(4-glycidyloxybenzoate) with 100 mL (1.14 mol) methacrylic acid in the presence of 0.3 g (1.14 mmol) of triphenyl phosphine as the catalyst. MEHQ (10 mg, 0.81 mmol) and Phenothiazine (10 mg, 50 µmol) were added as inhibitors and stabilizers. The reaction was run at 76 °C for up to 24 h. Excess methacrylic acid is first removed under vacuum and then the crude is mixed with DCM and washed with a saturated sodium carbonate solution. The final product is purified through column chromatography (ethyl acetate/hexanes-70:30 *w*/*w*) and stabilized with 500 ppm of MeHQ. The synthesis and characterization of intermediate compounds are provided in the Appendix A.

ISB4GBMA (69% yield, >98% pure), ^1^H NMR (CDCl_3_-δ, ppm): 1.9 (6H, 2x-CH_3_ methacrylate), 3.9–4.4 (14H, 4H-CH_2_ isosorbide, 8H-CH_2_, 2H-CH glyceryloxy), 5.03 (1H, -CH isosorbide), 4.6 (1H, -CH isosorbide), 5.39 (1H, -CH isosorbide), 5.45 (1H, -CH isosorbide), 5.6 (2H, =CH_2_ methacrylate), 6.15 (2H, =CH_2_ methacrylate), 6.9 (4H, -CH aromatic), 7.9 (2H, -CH aromatic), 8.04 (2H, -CH aromatic).

ISB3GBMA (76% yield, >98% pure), ^1^H NMR (CDCl_3_-δ, ppm): 1.9(6H, 2x-CH_3_ methacrylate), 3.9–4.4 (14H, 4H-CH_2_ isosorbide, 8H-CH_2_, 2H-CH glyceryloxy), 5.03 (1H, -CH isosorbide), 4.6 (1H, -CH isosorbide), 5.39 (1H, -CH isosorbide), 5.45 (1H, -CH isosorbide), 5.6 (2H, =CH_2_ methacrylate), 6.15 (2H, =CH_2_ methacrylate), 7.1 (2H, -CH aromatic), 7.3–7.4 (2H, -CH aromatic), 7.5 (1H, -CH aromatic), 7.6 (2H, -CH aromatic), 7.7 (1H, -CH aromatic).

ISB2GBMA (79% yield, >96% pure), ^1^H NMR (CDCl_3_-δ, ppm): 1.9(6H, 2x-CH_3_ methacrylate), 3.9–4.4 (14H, 2H-CH isosorbide, 2H-CH_2_ isosorbide, 8H-CH_2_, 2H-CH glyceryloxy), 5.03 (1H, -CH isosorbide), 4.6 (1H, -CH isosorbide), 5.39 (1H, -CH isosorbide), 5.45 (1H, -CH isosorbide), 5.6 (2H, =CH_2_ methacrylate), 6.15 (2H, =CH_2_ methacrylate), 6.9–7.0 (4H, -CH aromatic), 7.45–7.52 (2H, -CH aromatic), 7.8–7.9 (2H, -CH aromatic).

ISDGMA was synthesized according to the reaction scheme in Figure 5.

Isosorbide was etherified in the presence of allyl bromide, potassium hydroxide, and water. The subsequent reaction step with MCPBA provided isosorbide diglycidyl ether. ISDGMA was synthesized via the addition of 75 mL (0.85 mol) methacrylic acid to 20 g (77 mmol) of isosorbide diglycidyl ether in the presence of 0.15 g (0.57 mmol) triphenylphosphine. MEHQ (12.5 mg, 0.1 mmol) and Phenothiazine (12.5 mg, 63 µmol) were added as inhibitors and stabilizers. The reaction was run at 76 °C for up to 24 h. Excess methacrylic acid was first removed under vacuum and then the crude was mixed with DCM and washed with a saturated sodium carbonate solution. The final product was purified through column chromatography (ethyl acetate/hexanes-95:5 *w*/*w*) and stabilized with 500 ppm MeHQ. The synthesis and characterization of intermediate compounds are provided in the Appendix A.

ISDGMA (82% yield, >85% disubstituted with up to 15 mole% branching), ^1^H NMR (CDCl_3_-δ, ppm): 1.9 (6–7.5H, 2-CH_3_ methacrylate), 3.5–4.2 (16–20H, 4H-CH_2_ isosorbide, 2H-CH isosorbide, 8H-CH_2_, 2H-CH glyceryloxy), 4.5 (1H, -CH isosorbide), 4.65 (1H, -CH isosorbide), 5.6 (2–2.5H, =CH_2_ methacrylate), 6.15 (2–2.5H, =CH_2_ methacrylate). The varying proton area integration observed in various regions of the NMR spectrum will be addressed in the discussion section.

### 2.2. Resin Preparation and Evaluation

Resins were prepared by mixing each isosorbide monomer, or the reference BisGMA material, with TEGDMA in a 60:40 weight ratio, and by adding the photoinitiator, camphorquinone, and the co-initiator, dimethyl aminoethyl methacrylate (0.4% and 1.0%, respectively, by weight) [27]. The final mixture was poured into a stainless-steel mold and bounded by two microscope slides. The mixture was then cured using an AZDENT 1900 mW/cm^2^ LED curing light with an 8 mm diameter light guide tip, operating in the wavelength range of 400–500 nm, and placed 50 mm above the mold. This polymerization technique was adopted as a general method to compare the different resins.

Resin mixtures for water sorption testing had the dimensions of (1 mm thick × 15 mm diameter) and were cured for 40 s on the top and bottom sides. ISDGMA/TEGDMA samples were cured for 80 s on both sides for reasons to be addressed in the discussion section.

The evaluation for the remaining tests (degree of conversion, polymerization shrinkage, glass transition temperature, flexural strength and modulus) was done on resin mixture samples with 2 mm × 6 mm × 38 mm dimensions, in which the top and bottom sides were cured for 40 s twice in an alternating manner. The resulting samples were evaluated as is. For the degree of conversion and polymerization shrinkage, five samples were evaluated, while the evaluation of flexural strength and glass transition temperatures was done on three samples.

Water sorption: Five disc-shaped samples were prepared to measure the water sorption according to ISO4049 [35]. Upon curing, the samples were placed in a 37 °C oven to obtain a constant mass. Then, they were placed upright in the center of a 14 mL vial using a holder, and 10 mL of DPBS were added. The samples were kept at 37 °C for 7 days. Samples were then removed and pat dried with Kim wipes before mass measurement. Water sorption was measured according to Equation (1), where m_7_ is the weight after 7 days of immersion, m_0_ is the initial weight prior to immersion, and *V*_0_ is the initial volume of the specimen.
(1)$$\mathrm{Water}\;\mathrm{sorption}\;(\mathrm{WS},\;µg/{\mathrm{mm}}^{3})=\frac{m_{7}-m_{0}}{V_{0}}$$

Degree of monomer conversion: ATR-FTIR was used to measure the degree of conversion of the methacrylate double bonds according to Equation (2) in polymers derived from BisGMA and ISBGBMA monomers [36]. The degree of conversion was calculated using the polymer to monomer ratio of the absorbance height of the methacrylate vinyl C=C stretching at a frequency ν = 1636 cm^−1^. The spectra of the absorbance bands were normalized using the C=C aromatic stretching band at ν = 1610 cm^−1^ as an internal standard. The degree of conversion is reported as the average between the top and bottom layers of the test specimen. On the other hand, ISDGMA does not contain an aromatic group to be used as an internal standard, and as such, its degree of conversion was not calculated in this study.
(2)$$\mathrm{Degree}\;\mathrm{of}\;\mathrm{conversion}=\left[1-\frac{A\left(c=c\right)\;methacrylate\;\left(polymer\right)}{A\left(c=c\right)\;methacrylate\;\left(monomer\right)}\right]\times 100\%$$

Polymerization shrinkage: A 25 mL pycnometer (Wilmad lab glass, Vineland, NJ, USA), and an analytical balance (OHAUS explorer, Parsippany, NJ, USA), were used to determine the mass, volume, and the corresponding densities of the cured specimens. Polymerization shrinkage, [PS(%)], as a result of volumetric shrinkage, was calculated according to Equation (3), where *d_p_* is the density of the polymer and *d_m_* is the density of the monomer mixture [37].
(3)$$\mathrm{PS}\;(\%)=\frac{d_{p}-d_{m}}{d_{p}}\times 100\%$$

Glass transition temperature: Glass transition temperature (T_g_) was determined using a three-point bending test in the temperature range of −40 °C to 200 °C, at a rate of 10 °C/min and a frequency of 1 Hz. T_g_ was evaluated at the maximum tan delta peak [38].

Flexural strength and modulus: Flexural strength was calculated based on a modified ISO 4049 method [35]. TA-XT Plus texture analyzer equipped with a 50 Kg load cell and a strain rate of 1.2 mm/min was used. Resin sample sets were divided into two groups. One group was measured after immersion in a DPBS buffer solution for 24 h at 37 °C, while the other group was measured after being placed in an oven at 37 °C for 24 h. Flexural strength (σ, MPa) was calculated using Equation (4), where F is the maximum load exerted on the specimen at the point of fracture (N), l is the distance between the supports (mm), b is the width of the specimen (mm), and h is the thickness of the specimen (mm). Flexural modulus was obtained from the slope in the linear region between 0.05–0.25% strain, similar to ISO 178 [39].
(4)$$\sigma =\frac{3Fl}{2bh^{2}}$$

## 3. Results and Discussion

Isosorbide with its two-ring structure and hydroxyl functionality can resemble Bisphenol A, and as such, it is typically viewed as a potential BPA replacement [40]. In the work of Łukaszczyk et al., and Shin and coworkers, ISDGMA was proposed as a BisGMA replacement [27,28]. However, because of its hydrophilic nature, ISDGMA is reported to have higher water sorption relative to BisGMA. To overcome this, three isosorbide dimethacrylate monomers with hydrophobic benzoate aromatic spacers were synthesized and evaluated. They differ amongst each other based on the position of the dimethacrylate substituent on the aromatic ring (para-, meta-, or ortho-).

Inspection of the ^1^H NMR spectra reveals that, during the final step of the synthesis of ISB4GBMA, the epoxide protons of the glycidyloxybenzoate intermediate at 3.37, 2.93, and 2.78 ppm disappear and give rise to methacrylate protons at 5.6 and 6.15 ppm, as shown in Figure 6 and Figure 7. Similarly, the related proton NMR spectra of the meta and ortho benzoate substituted isosorbides are provided in Appendix A.

Figure 8 displays the NMR spectrum of ISDGMA. Branching in the form of bis-methacrylation, up to 15 mole %, at the glyceryloxy hydroxyl functionality is suspected as evidenced by the excess area integration of the isosorbide cycle protons H3/H4 to the methacrylate protons at 5.6/6.1 ppm, and to the methyl protons at 1.9 ppm. The new ISBGBMA monomers were found to be less susceptible to such branching. The NMR spectrum of isosorbide diglycidyl ether is provided in Appendix A.

The benzoate functionality in ISBGBMA monomers was added to increase the hydrophobicity over ISDGMA. Contact angle measurements and cLogP estimations of ISBGBMA monomers, ISDGMA, and BisGMA are reported in Table 1. The mean contact angle (θ) of ISBGBMA monomers at (~62.9°) is higher than 13.41° for ISDGMA. This indicates that the new monomers are more hydrophobic [41]. However, they are less hydrophobic than BisGMA at 77°. The cLogP estimations indicate the ISBGBMA monomers to be more hydrophobic than ISDGMA, but less hydrophobic than BisGMA as well [42].

The viscosity of the monomer mixture is an important parameter that affects the degree of polymerization conversion and the amount of filler that can be incorporated into a composite. Therefore, it is typically preferred to have resin mixtures with lower viscosities as long as polymerization shrinkage and water sorption are kept to a minimum [43,44]. Viscosity increased for the ISBGBMA/TEGDMA monomer mixtures in the order of ortho < meta < para, as shown in Table 2. ISB4GBMA/TEGDMA had the highest viscosity at 2.49 Pa·s, while the ortho mixture ISB2GBMA/TEGDMA had the lowest at 0.63 Pa·s. The viscosity of the ortho mixture was near to that of the BisGMA/TEGDMA reference at 0.48 Pa·s and was noticeably higher than the viscosity of ISDGMA/TEGDMA at 0.06 Pa·s. In the work of Kim et al., the viscosity of ISDGMA/TEGDMA at 70:30 wt% was similar to that of BisGMA/TEGDMA at 60:40 wt% [28].

The size and shape of the molecule, its molecular weight, and ability for inter and intramolecular interactions have an impact on its viscosity [7]. The ISBGBMA monomers are larger at (~671 g/mol) in comparison to ISDGMA (~431 g/mol) and BisGMA (~513 g/mol), and can therefore cause more friction and increase the viscosity of the mixture. On the other hand, since all the ISBGBMA monomers have the same molecular weight, we propose that the difference in viscosity is likely related to the conformation of the isosorbide monomer. ISB4GBMA is relatively planar while ISB2GBMA is sterically hindered; planar aromatic molecules are known to closely align in parallel, in what is called π-stacking [45]. The steric hindrance and lower potential of ISB2GBMA for packing is believed to result in lower friction and lower viscosity for ISB2GBMA/TEGDMA.

The degree of conversion, polymerization shrinkage, and glass transition temperatures are presented in Table 3. Conversion of p(ISB2GBMA/TEGDMA) at 52% was significantly different from the conversion of p(ISB4GBMA/TEGDMA) at 47%, but statistically similar to the conversion of p(BisGMA/TEGDMA) at 54%. As stated earlier, ISDGMA does not contain an aromatic ring and its degree of conversion was not calculated in this study. On the other hand, Łukaszczyk et al., reported the degree of conversion of p(ISDGMA/TEGDMA) at 60:40 wt% to be slightly higher than that of P(BisGMA/TEGDMA) by means of photo-DSC in their studies [27].

Unreacted double bonds can indicate the presence of free monomers or pendant groups. A degree of conversion of less than 50% suggests the presence of unreacted residual dimethacrylates. Free monomers can leach and irritate the soft tissue, while pendant groups lower chemical crosslink density and reduce mechanical integrity [6,46]. The efficiency of the photopolymerization system and the stiffness and elasticity of the monomers can affect the degree of conversion [47]. The addition of a diluent helps increase the degree of conversion through a reduction in the overall viscosity and an increase in reaction diffusion [48].

In the work of Pfeifer et al., the degree of conversion of p(BisGMA) was improved with increasing TEGDMA content and lowering the viscosity of the mixture [49]. Therefore, the higher degree of monomer conversion of p(ISB2GBMA/TEGDMA), in comparison to the other ISBGBMA isomers, is attributed to its lower viscosity.

The polymerization shrinkage was highest around 7% for p(ISB2GBMA/TEGDMA), p(BisGMA/TEGDMA), and p(ISDGMA/TEGDMA) and was statistically different from the polymerization shrinkage of p(ISB4GBMA/TEGDMA) at 4.25%. Polymer systems shrink as the covalently bonded monomers occupy less space [37]. This depends on the degree of conversion, functionality, and molecular weight [50]. Polymerization shrinkage causes stresses within the matrix and within the matrix and tooth interface. It is associated with microleakage, gap formation, enamel crack propagation, and post-operative sensitivity [51].

The lower degree of conversion for the para- and its bulky structure is believed to result in lower polymerization shrinkage. According to Pfeifer et al. [49], higher degrees of conversion resulted in higher polymerization shrinkage in p(BisGMA/TEGDMA). Similarly, the polymerization shrinkage of p(ISDGMA/TEGDMA) was slightly higher than that of p(BisGMA/TEGDMA) in the work of Łukaszczyk et al., and was attributed to higher degrees of monomeric conversion [27].

Glass transition temperatures were determined from the maxima of tan delta curves, as shown in Figure 9.

The glass transition temperature must exceed cure temperature and temperature ranges encountered in the oral environment, for the material to be clinically viable [7]. The lower T_g_ of the polymers derived from ISBGBMA, relative to that of p(BisGMA/TEGDMA), are attributed to their relative lower degrees of monomeric conversion. Stansbury et al. and Sideridou and co-workers demonstrated higher glass transition temperatures that were obtained with higher degrees of conversion [6,7]. The polymer derived from ISB2GBMA had a slightly higher glass transition temperature (86 °C) than the polymer derived from ISB4GBMA (80 °C) or ISB3GBMA (77 °C). Steric hindrance, increased rigidity, and improved degree of conversion are likely to be the cause for this behavior.

In contrast, statistical analysis did not show a significant difference between the glass transition temperatures of p(BisGMA/TEGDMA) at 95 °C and p(ISB2GBMA/TEGDMA) at 86 °C, suggesting the corresponding crosslinked networks to be structurally similar. The glass transition temperature of p(ISDGMA/TEGDMA) at 85 °C was statistically different only from the T_g_ of p(BisGMA/TEGDMA). On the other hand, the glass transition temperature of p(BisGMA/TEGDMA) and p(ISDGMA/TEGDMA) were reported to be comparable in the work of Łukaszczyk et al., and were attributed to similar degrees of conversion between the two polymers. Different curing and Tg characterization methods were used in that study [27].

The corresponding storage modulus is shown in Figure 10. It is highest for p(BisGMA/TEGDMA) and p(ISB2GBMA/TEGDMA) where the degree of conversion is higher. The low modulus obtained for p(ISDGMA/TEGDMA) might be attributed to its structural flexibility.

The lower water sorption of p(ISB2GBMA/TEGDMA) at 39 µg/mm^3^ was statistically different from the water sorption of p(ISB4GBMA/TEGDMA) at 44 g/mm^3^. However, the polymer derived from BisGMA had the lowest water sorption at 26 µg/mm^3^, and the polymer derived from ISDGMA had the highest at 73 µg/mm^3^, as shown in Figure 11. 

The improved hydrophobicity of the ISBGBMA series over ISDGMA helped lower the water sorption by about 40%. However, it was not enough to result in lower water sorption than p(BisGMA/TEGDMA). The isosorbide core is hygroscopic with hydrophilicity similar to diethylene glycol [20,52]. Therefore, improving the hydrophobicity further is expected to reduce the water sorption further.

Hygroscopic expansion due to water can weaken mechanical properties, increase wear rate, and reduce dimensional stability [53,54,55]. Polymer samples derived from the hydrophilic ISDGMA deteriorated and cracked as a result of water uptake after immersion in buffer solution, as shown in Figure 12a. This was also noted by Kim et al., where p(ISDGMA) resins and resin-based composites showed surface cracks after buffer storage [33]. On the other hand, polymer samples derived from the ISBGBMA monomers exhibited greater stability due to improved hydrophobicity and reduced water sorption, similar to the BisGMA reference. This is illustrated in Figure 12b,c.

When ISDGMA/TEGDMA samples were initially cured similarly to other resin samples for water sorption, the results obtained were inconsistent. Lower degree of conversion, sample deterioration, and dissolution are probable causes. To optimize the curing condition further, the samples were cured longer. As a result, the water sorption was consistent, and this is shown in Appendix A.

In addition, the choice to store the samples in DPBS instead of DI water for water sorption was to compare this data to ISDGMA values reported in the literature [27,28]. To determine any interference that the buffer solution may have to the water sorption data, SEM-EDS was used to determine the extent of phosphorus deposition on all sample groups. The results revealed no detectable deposition levels and therefore, any interference to the water sorption from the storage medium would be negligible.

Dental restorative materials should have sufficient mechanical integrity to function properly. While ISO 4049 reports on the minimum requirements for resin-based composites, it is generally recognized that the higher the mechanical strength is, the better is the performance [56]. In our studies, flexural strength analysis showed a decrease in strength for all samples once immersed in DPBS for 24 h as shown in Figure 13. The decrease in strength was greatest (80%) for p(ISDGMA/TEGDMA), where signs of network degradation were observed. All other samples had a comparable 50% loss.

Although one can note differences in flexural strength across various groups as indicated in the graph, statistical analysis did not confirm such variations, which might be due to the limited number of sample sets per group. The flexural strength of p(BisGMA/TEGDMA) was statistically significant only from p(ISDGMA/TEGDMA) in both dried and immersed states. On the other hand, p(ISDGMA/TEGDMA) was statistically significant from all samples in the immersed sample set.

In the work of Yiu et al., the tensile strength of unfilled dental resins was lower for hydrophilic materials upon water storage, where water sorption was highest and water acted as a plasticizer [57]. As such, we believe the improved hydrophobicity and increased rigidity of the ISBGBMA monomers over ISDGMA, through the presence of the aromatic groups, to improve flexural strength by reducing the water uptake of the samples. Table 4 shows the percent water uptake with respect to mass after 24 h of sample storage. The hydrophilic material p(ISDGMA/TEGDMA) had the highest water uptake and the lowest flexural strength. Since the storage time was short (24 h), it is unclear if saturation was obtained, and thus, caused such significant decreases in flexural strength. Longer storage times are needed to verify this concept further. Similarly, Kim et al. reported a decrease in flexural strength in resins and resin-based composites of p(BisGMA/TEGDMA) and p(ISDGMA) after PBS buffer storage for 7 days at 37 °C [33].

The lower flexural strength of the ISBGBMA polymers, in comparison to p(BisGMA/TEGDMA), is due to the lower degree of monomeric conversion and corresponding cross link density, where the concentration of methacrylate double bonds is 5.13 mol/Kg for BisGMA/TEGDMA, and 4.58 mol/Kg for ISBGBMA/TEGDMA. Gajewski et al. have shown that flexural strength improves with higher degree of conversion and cross link density [58].

The flexural modulus is given in Figure 14; it is highest for p(BisGMA/TEGDMA) and p(ISB4GBMA/TEDMA) in the samples dried at 37 °C. However, no statistical significance in the modulus was noted between all samples dried at 37 °C. As expected, the flexural modulus of the immersed samples is lower. In the work of Ito et al., the modulus of elasticity was lower in resins with higher water sorption. The more hydrophilic the resin is, the higher is the water sorption, and the lower is the modulus of elasticity [59]. The flexural modulus of p(ISDGMA/TEGDMA) was statistically significant from all samples in the immersed state. Similar to flexural strength, the limited number of sample sets per group may contribute to the findings of statistical analysis.

Additionally, the reported modulus of p(BisGMA/TEGDMA) in this study might be lower than what is described in the literature [60,61]. We attribute these differences to the polymerization technique in this work, which as stated earlier, is adopted as a general approach to compare the different resins. Thus, the modulus and the corresponding properties of all polymers reported herein, are dependent on the polymerization condition, where light intensity at the sample surface and various depths might play a role.

## 4. Conclusions

The focus of this study was to identify bio-based, BPA-free, dental resin methacrylates with similar key characteristics to BisGMA. In this regard, three hydrophobically-modified isosorbide dimethacrylate isomers (ISBGBMA) were synthesized, characterized, and evaluated in the form of copolymers with TEGDMA as potential dental filling resins. ISBGBMA monomers were more hydrophobic than ISDGMA based on contact angle measurements and cLogP estimations, but less hydrophobic than BisGMA.

The para, meta, and ortho substitution of the benzoate functionality in the ISBGBMA monomers resulted in different properties. In this series, the monomer mixture of TEGDMA and the ortho ISBGBMA monomer had the least viscosity. The corresponding copolymer had the lowest water sorption and the highest degree of monomeric conversion, polymerization shrinkage, glass transition temperature, and storage modulus. However, flexural strength was comparable, while the flexural modulus was highest for the copolymer of TEGDMA and the para-substituted ISBGBMA monomer in the dry sample set.

The hydrophobic ISBGBMA monomers showed greater improvement over the hydrophilic ISDGMA monomer. The corresponding copolymers with TEGDMA had lower water sorption and improved mechanical strength, modulus, and buffer stability. On the other hand, the viscosity of their monomeric mixtures with TEGDMA was much higher.

The performance of the ortho ISBGBMA monomer was comparable to that of BisGMA among the ISBGBMA series. The viscosity of the monomer mixture with TEGDMA was similar. The degree of conversion of the corresponding copolymer, polymerization shrinkage, glass transition temperature, storage modulus, flexural strength, and modulus resembled those of p(BisGMA/TEGDMA). However, the water sorption was significantly higher.

Taken together, these data suggest that the ortho ISBGBMA monomer is a potential bio-based, BPA-free replacement for BisGMA, and should be the focus for future studies.

## Figures and Tables

**Figure 1 materials-14-02139-f001:**
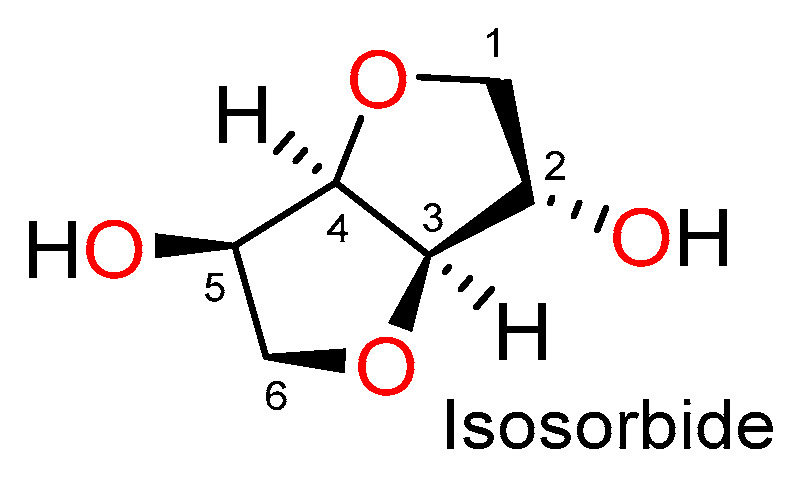
Chemical structure of isosorbide.

**Figure 2 materials-14-02139-f002:**
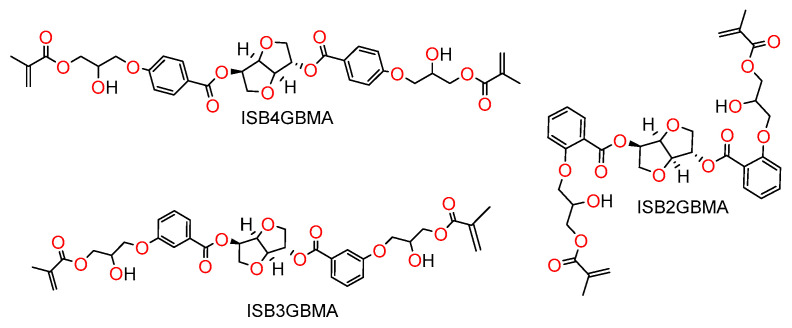
Hydrophobically modified isosorbide dimethacrylates with a para, meta, and ortho aromatic spacer.

**Figure 3 materials-14-02139-f003:**
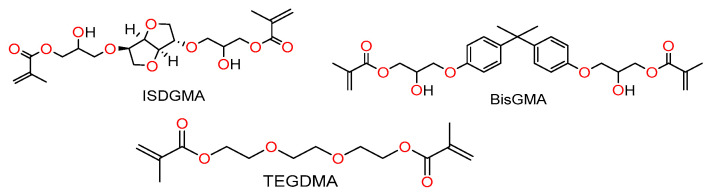
ISDGMA (1,4:3,6-dianhydro-d-sorbitol 2,5-bis(2-hydroxy-3-methacylolxypropoxy)), BisGMA (2,2-bis [4-(2-hydroxy-3-methacryloyloxypropoxy)phenyl] propane), and TEGDMA (triethylene glycol dimethacrylate).

**Figure 4 materials-14-02139-f004:**
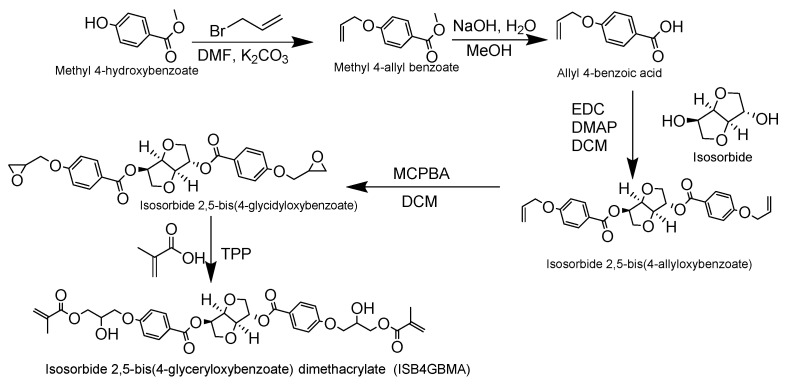
Reaction scheme for ISB4GBMA (1,4:3,6-dianhydro-d-sorbitol 2,5-bis [4-(2-hydroxy-3-methacryloyloxypropoxy)benzoate]).

**Figure 5 materials-14-02139-f005:**
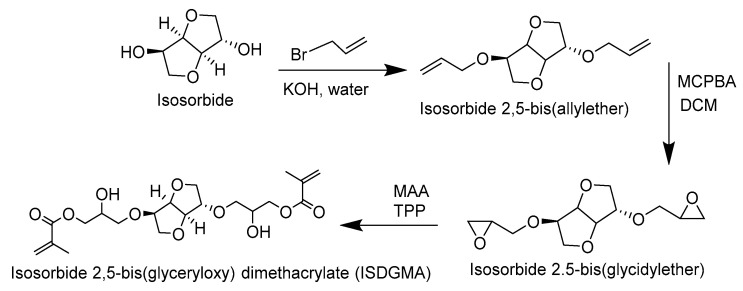
Reaction scheme for ISDGMA.

**Figure 6 materials-14-02139-f006:**
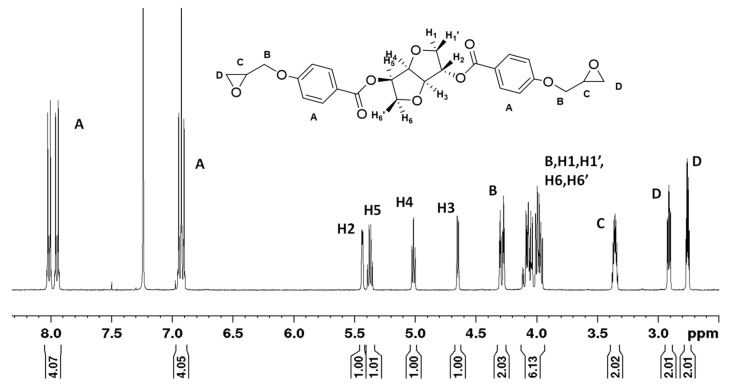
^1^H NMR Isosorbide 2,5-bis(4-glycidyloxybenzoate).

**Figure 7 materials-14-02139-f007:**
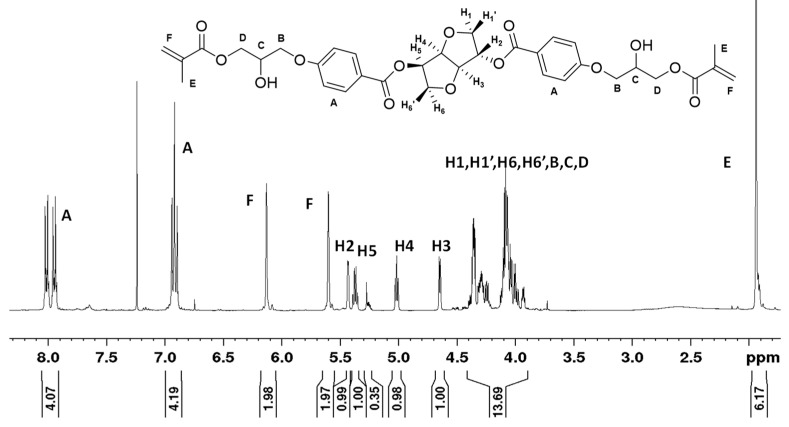
^1^H NMR Isosorbide 2,5-bis(4-glycerloxybenzoate) dimethacrylate.

**Figure 8 materials-14-02139-f008:**
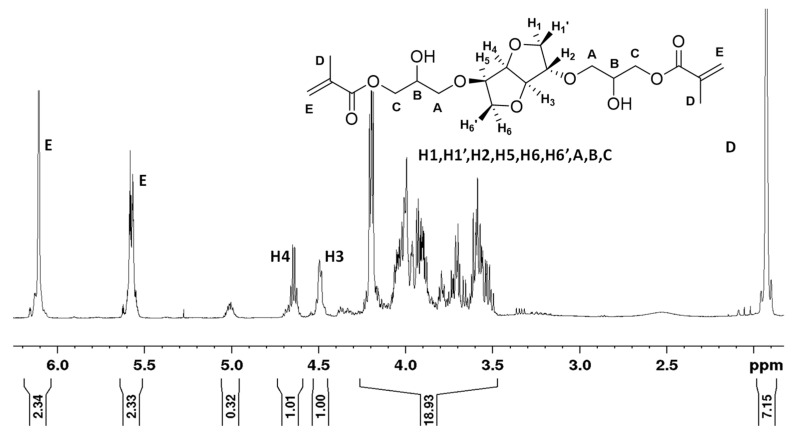
^1^H Isosorbide 2,5-bis(glyceryloxy) dimethacrylate.

**Figure 9 materials-14-02139-f009:**
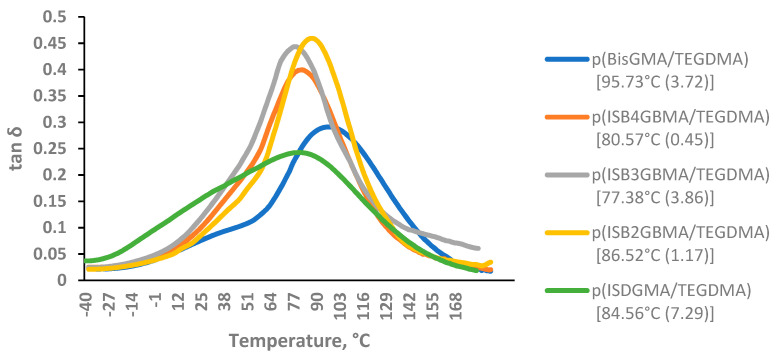
Tan delta of p(BisGMA/TEGDMA), p(ISB4GBMA/TEGDMA), p(ISB3GBMA/TEGDMA), p(ISB2GBMA/TEGDMA), and p(ISDGMA/TEGDMA) at 60:40 wt%. Maximum tan delta value in brackets and standard deviation in parenthesis.

**Figure 10 materials-14-02139-f010:**
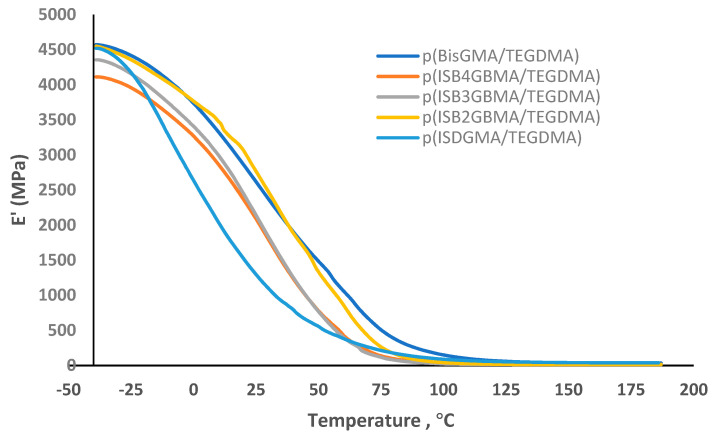
Storage modulus of p(BisGMA/TEGDMA), p(ISB4GBMA/TEGDMA), p(ISB3GBMA/TEGDMA), p(ISB2GBMA/TEGDMA), and p(ISDGMA/TEGDMA) at 60:40 wt%.

**Figure 11 materials-14-02139-f011:**
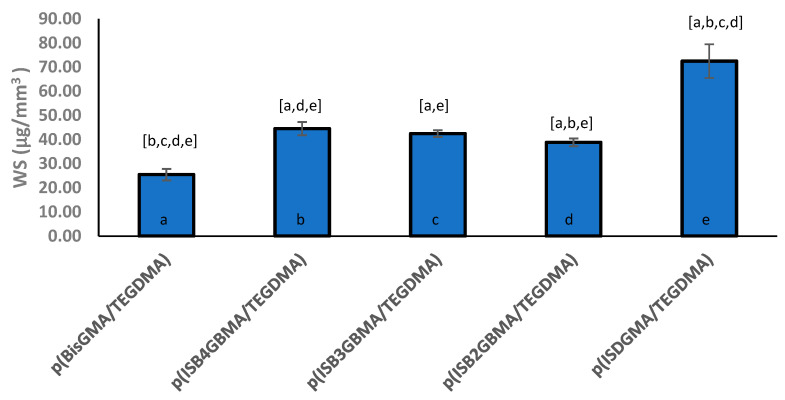
Water sorption of p(BisGMA/TEGDMA), p(ISB4GBMA/TEGDMA), p(ISB3GBMA/TEGDMA), p(ISB2GBMA/TEGDMA), and p(ISDGMA/TEGDMA) at 60:40 wt%. Letters indicate statistically significant difference (*p* < 0.05) based on group. Individual groups are represented by (a–e).

**Figure 12 materials-14-02139-f012:**
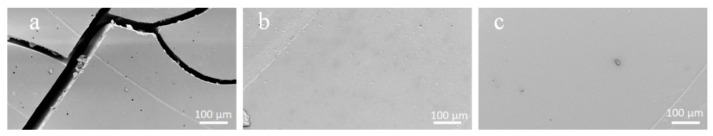
SEM images at 150× and 2.5 kV of (**a**) p(ISDGMA/TEGDMA), (**b**) p(ISB2GBMA/TEGDMA), and (**c**) p(BisGMA/TEGDMA) after buffer immersion for 7 days at 37 °C.

**Figure 13 materials-14-02139-f013:**
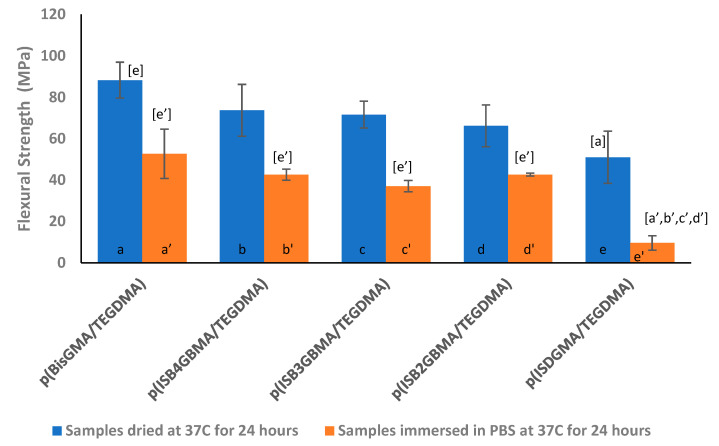
Flexural strength of p(BisGMA/TEGDMA), p(ISB4GBMA/TEGDMA), p(ISB3GBMA/TEGDMA), p(ISB2GBMA/TEGDMA), and p(ISDGMA/TEGDMA) at 60:40 wt% at RT. Letters indicate statistically significant difference (*p* < 0.05) based on group.

**Figure 14 materials-14-02139-f014:**
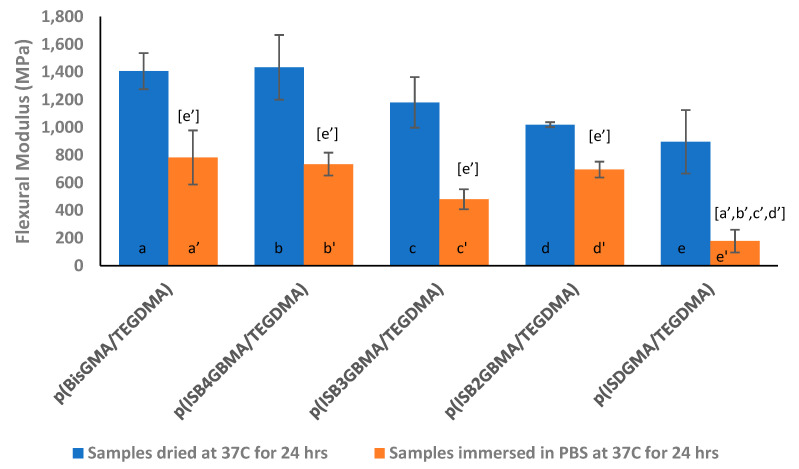
Flexural modulus of p(BisGMA/TEGDMA), p(ISB4GBMA/TEGDMA), p(ISB3GBMA/TEGDMA), p(ISB2GBMA/TEGDMA), and p(ISDGMA/TEGDMA) at 60:40 wt% at RT. Letters indicate statistically significant difference (*p* < 0.05) based on group.

**Table 1 materials-14-02139-t001:** Contact Angle Measurements and cLogP Estimations of BisGMA, ISB4GBMA, ISB3GBMA, ISB2GBMA, and ISDGMA.

Sample	Contact Angle (Degrees °)	cLogP
BisGMA ^a^	76.97 (2.84) ^[b,c,d,e]^	5.09
ISB4GBMA ^b^	63.21 (4.08) ^[a,e]^	2.75
ISB3GBMA ^c^	62.65 (1.33) ^[a,e]^	2.75
ISB2GBMA ^d^	62.82 (1.74) ^[a,e]^	2.75
ISDGMA ^e^	13.41 (1.76) ^[a,b,c,d]^	−0.53

Letters indicate statistically significant difference (*p* < 0.05) based on group. Standard deviation in parenthesis.

**Table 2 materials-14-02139-t002:** Brookfield Viscosity of BisGMA/TEGDMA, ISB4GBMA/TEGDMA, ISB3GBMA/TEGDMA, ISB2GBMA/TEGDMA, and ISDGMA/TEGDMA at 60:40 wt% and 25 °C.

Sample	Viscosity at 25 °C (Pa·s)
BisGMA/TEGDMA	0.48
ISB4GBMA/TEGDMA	2.48
ISB3GBMA/TEGDMA	1.26
ISB2GBMA/TEGDMA	0.63
ISDGMA/TEGDMA	0.06

**Table 3 materials-14-02139-t003:** Degree of Conversion, Polymerization Shrinkage, and Glass transition Temperature of p(BisGMA/TEGDMA), p(ISB4GBMA/TEGDMA), p(ISB3GBMA/TEGDMA), p(ISB2GBMA/TEGDMA), and p(ISDGMA/TEGDMA) at 60:40 wt%.

Sample	Degree of Conversion (%)	Polymerization Shrinkage (%)	Glass Transition Temperature (°C)
p(BisGMA/TEGDMA) ^a^	54 (3) ^[b]^	6.54 (1) ^[b]^	95.73 (3.73) ^[b,c,e]^
p(ISB4GBMA/TEGDMA) ^b^	47 (4) ^[a,d]^	4.25 (1) ^[a,d,e]^	80.57 (0.46) ^[a]^
p(ISB3GBMA/TEGDMA) ^c^	51 (3)	5.41 (1) ^[d,e]^	77.38 (3.86) ^[a]^
p(ISB2GBMA/TEGDMA) ^d^	52 (5) ^[b]^	6.7 (1) ^[b,c]^	86.52 (1.18)
p(ISDGMA/TEGDMA) ^e^	N/A	7.28 (1) ^[b,c]^	85.56 (7.30) ^[a]^

Letters indicate statistically significant difference (*p* < 0.05) based on group. Standard deviation in parenthesis.

**Table 4 materials-14-02139-t004:** Water uptake after 24 h and at 37 °C in DPBS for p(BisGMA/TEGDMA), p(ISB4GBMA/TEGDMA), p(ISB3GBMA/TEGDMA), p(ISB2GBMA/TEGDMA) and p(ISDGMA/TEGDMA).

Test	p(BisGMA/TEGDMA)	p(ISB4GBMA/TEGDMA)	p(ISB3GBMA/TEGDMA)	p(ISB2GBMA/TEGDMA)	p(ISDGMA/TEGDMA)
WU%	0.91% (0.05)	1.48% (0.09)	1.35% (0.13)	1.19% (0.06)	4.20% (0.17)

WU%: water uptake percentage. Standard deviation in parenthesis.

## Data Availability

All data presented in this study is available in Marie, B; Clark, R; Gillece, T; Ozkan, S; Jaffe, M; Ravindra, N.M. Hydrophobically Modified Isosorbide Dimethacrylates as Bisphenol-A (BPA)-Free Dental Filling Materials.

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
