# Peer review of "Hydrophobically Modified Isosorbide Dimethacrylates as a Bisphenol-A (BPA)-Free Dental Filling Material"

_materials, 2021, doi:10.3390/ma14092139_

Round 1

Reviewer 1 Report

The manuscript was corrected according to my comments.

Reviewer 2 Report

Dear authors,

Unfortunately, the resubmitted version of your article still needs minor changes:

  • prepare tables according to the template
  • prepare de references list according to the template
  • section 3, 4 should be part of m&m, and numbered as 2.1, 2.2 etc
  • section 5, results and discussion should become section 3 etc

Reviewer 3 Report

This article is resubmitted version of the initial work by the same title. In previous version word has serious flaws so it was rejected by the editor. So in this road I feel obligated to check if suggested revision has been done in proper way and next rate the work again.

A number of changes have been made to the work, but I believe most of them have been implemented in a too modest way. Moreover, some of the changes introduced confused the reviewer.

Old problems (with some new aspects):

  1. “Why the phosphate buffer was used as storing medium? This is not a popular approach for these materials.” – this is still not discussed. Moreover, authors kept the samples either in the oven or in PBS (both 37°C). The time was short, 24 hours (FS, E) or 7 days (WS) -it is therefore surprising that they did not provide any explanation as to why, after such a short time, PBS caused such a significant decrease in mechanical properties???
  2. “For some tests, sample dimensions were not given.” - This was modified, but I still have some doubts. For example, why for flexural strength test 2 mm × 6 mmx 38 mm samples were prepared, when the ISO require 2x2x25? Moreover, you wrote that “the top and bottom sides were cured for 40 seconds twice in an alternating manner” but how it was possible when you used “8 mm diameter light guide tip”? As we can see, this description is still unreliable prepared. Number of samples in each group were not mentioned.

  1. “Resin preparation and evaluation – for this section citation are strongly needed.” - The authors refer to literature in some, but suggest introducing modifications (e.g. in comparison to standards) without specifying their scope. Moreover, methods of the samples surfaces preparation have not been described, which in many cases significantly affects the test results – If surfaces were not finished, I should be clearly mentioned.
  2. “The results of statistical analyzes have not been presented and there is nothing about it in the methodology. Some new results are still without statistical analysis - Contact Angle....
  3. „The discussion is shallow. (...)”. - No major changes have been made in this regard. The discussion is still at a low level - it is rather in the form of an author's commentary. The results were not discussed against the background of the literature.
  4. “You have tested water sorption in PBS - what are the guarantees that the result was not affected by the deposition of PBS components on the surface of the samples during the 7 days of exposure?”

“Explain why the sorption was lower after 7 days than after 1 day? This is incomprehensible. Why the research was during 7 days was carried out for only one material?”

This has not been clarified and not been corrected. Moreover, the authors simply removed some of the solubility results (water absorption after 1 day) that aroused the consideration of the reviewer. This is not the right way to clear up methodological doubts.

Some new noticed problems:

  1. The composition of bisGMA/TEGDMA is well known, so and well described in the literature (including similar compositions). It is therefore surprising that the authors did not notice and comment in the discussion that the E module in their work was at least twice lower than that usually found for this type of material (other similar acrylates). This module is even lower than for PA ... Could that be the reason why the value of E was not shown in the first version of the paper?

Please see:

Xiaoyan Wang et all. Improved performance of Bis-GMA/TEGDMA dental composites by net-like structures formed from SiO2 nanofiber fillers. Materials Science and Engineering C

Comparative Study of Structure-Property Relationships in Polymer Networks Based on Bis-GMA, TEGDMA and Various Urethane-Dimethacrylates by Izabela Barszczewska-Rybarek andSebastian Jurczyk Materials 2015, 8(3), 1230-1248; https://doi.org/10.3390/ma8031230 - 19 Mar 2015

  1. For dried samples FS (most groups) and E (all) you didn’t note significant differences, however we can see that differences are really large – did you checked power of the tests? How many samples per group were used?

In conclusion, I believe that the authors made corrections, but they are insufficient.

Reviewer 4 Report

 Now this paper can be published

Reviewer 5 Report

The paper describes the synthesis and the photopolymerization of bis-méthacrylates based on isosorbide rather than bisphenol A, followed by the characterization of the corresponding polymers in view of dental applications. The paper discusses the relation between the chemical structure of the bis-methacrylate and the properties of the corresponding polymer to fit with the requirements of the dental application. The paper is well-written, has clear objectives and brings important results for the preparation of bisphenol A-free polymers for dental applications. I recommend this paper for publication.

I have just one major remark regarding the experimental section describing the synthesis and the characterization of all the intermediate products, which is too brief. It is essential to describe the detailed synthetic procedures for all products, the yields of all reactions and the characterization of all products in the same way this is done in all journal on organic chemistry. This is an important information to help to the chemists repeating the synthesis of these products in their own laboratories. In the present version, the paper describes the detailed characterization of the final bis-methacrylates but not of the intermediated prepared for their synthesis. Of course, if a synthesis is already reported and the corresponding products fully characterized elesewhere, a citation to this work is sufficient. Of course, a full description for all products is too long but it can be added in the supporting informations.

Reviewer 6 Report

The authors should discuss the points from the clinical perspective  regarding the use of BPA in the daily dental practice. For example side effects such as: reproductive, immunity, and neurological problems, Alzheimer, , childhood asthma, metabolic disease, type 2 diabetes, and cardiovascular disease.

The literature should be up-to-date. Recent studies should be comparatively discussed with the results of the study. 

Regards

Round 2

Reviewer 3 Report

I have carefully read the revised version of the work and the arguments presented by the authors. The authors explained or supplemented some of the information, but the presented explanations reassured the reviewer that some improvements in the text must be done before acceptance of the manuscript.

We should take into account that the researchers moved the polymerization lamp significantly away from the samples (50mm), which must have influenced the intensity of the light reaching the sample and the intensity of the light at different depths of the polymerized samples. There is no information as to whether any measures have been taken to prevent this phenomenon and to validate the proposed polymerization method. I believe that it was the polymerization method that could have influenced the low modulus of elasticity of the tested materials. However, given that the authors have supplemented the methodology, which means that the readers will not be misled, I am able to accept such an approach, but I would ask the authors to make a clear reference to this issue in the discussion, emphasizing that such a solution was adopted due to the fact that this is a comparative study of experimental resins. It must be clearly stated that this module has these properties achieved for the polymerization conditions specified by the authors, and in the case of increasing the light intensity or extending the exposure time, they may not be different.

Reviewer 6 Report

The requested issues were addressed from the authors.

Author Response

Authors are very thankful to the Reviewers for the very helpful suggestions.

This manuscript is a resubmission of an earlier submission. The following is a list of the peer review reports and author responses from that submission.

Round 1

Reviewer 1 Report

Thanks for the article and I have the following comments:

1. "However, it contains about 50%  mercury and releases small amounts of the toxic inorganic material into the oral environment" This is not correct. Dental amalgam has been used for many many years and there is no evidence for proving the silver amalgam is toxic. The fade-out of dental amalgam is due to the environmental aspects (i.e. pollution) of mercury or due to its production. There is a Minamata Convention on Mercury to reduce to use of mercury. Please change the sentence and reference.

2. Please change "composites" into "resin composites" throughout the whole text.

3. "Despite their widespread use, BisGMA and other dental dimethacrylates" But your study is also making new dimethacrylates! So, maybe you should mention explicitly what dimethacrylates are negatively affecting human health.

4. What is the diameter of the light guide of LED LCU? Does it larger than 15mm ? Why different samples used different curing time? The SS mould is a split mould or?

5. I do not understand "2 × 3x8 mm". Do you mean "2mm × 3mm x 8 mm"?

5. Since you are using a disc shaped specimen, so how do you put the sample in the water sorption test? What is the container? What kind of phosphate buffer that you have been used?  If you are using ISO 4049, have you straightly the kimwipes to pad dry the specimen before you measure the mass? All details are missing.

6. I was wondering how do you use a liquid pycnometer to measure the shrinkage.  What is the volume or mass? Or are you measure solely the weight? Can you give us more details? If you are following ISO 1675, you should at least mention the balance and also the glassware. In dentistry, mentioning the shrinkage due to mass is just useful for QC, not an very important parameter. Instead, particular points worth to note should be volumetric shrinkage, shrinkage stress and shrinkage strain. Fig 11 and the results are odd to dental applications.

7. DMA Q800 and TA-XT - what are the sizes of the specimen for 3 point bending?  In addition, Q800 and TA-XT all can give you the 3 point bending values - why do you need to use these two machines? For TA-XT, why 1.2mm/min rate is used? 24 hrs storage seems not ok.

8. Figs. 6-8 - the chemicals are poorly drawn! 

9. Figs. 9, 10 --> change into table form, and add the SD and proper statistics like one-way ANOVA.

10. Fig 12 - perhaps you should let us know the stress or load modulus vs temperature curves in addition to the tan δ vs temperature curves.

11. 7 days storage is not something new. I suggest you should do 14days and 30days to illustrate the long term trends.

12. Please compare the results of Fig 16 with  proper statistics. Can one-way ANOVA and Tukey post hoc test help?

13. There is no proof this is hydrophobic but you have mentioned this is hydrophobic. So, please add an experiment or use a reference to tell us.

Reviewer 2 Report

The work addresses interesting issues, but has many imperfections.

  • The aim and thesis of the work were not formulated.
  • “The benzoate functionality is added to increase the hydrophobicity” so I'm surprised that the materials have not been tested in this regard.
  • Why the phosphate buffer was used as storing medium? This is not a popular approach for these materials.
  • Resin preparation and evaluation – for this section citation are strongly needed.
  • For some tests, sample dimensions were not given.
  • The results of statistical analyzes have not been presented and there is nothing about it in the methodology. This seems strange as it is mentioned in the results description. It must necessarily be fulfilled.
  • The discussion is shallow. It has not even commented on the desired or minimum properties of the analyzed materials. The results were not discussed in the context of recommended properties fr this materials. For example, it is often considered that the conversion rate is 50% or more. The bending strength for some materials is also very low, etc. These matters have not been commented on at all. The clinical relevance of the particular properties also has not been commented.
  • You have tested water sorption in PBS - what are the guarantees that the result was not affected by the deposition of PBS components on the surface of the samples during the 7 days of exposure?
  • Explain why the sorption was lower after 7 days than after 1 day? This is incomprehensible. Why the research was during 7 days was carried out for only one material?
  • Why was the modulus of elasticity not calculated - this is an important property of these materials?
  • The conclusions do not adequately reflect the results obtained and, in my opinion, do not properly reflect the situation

Reviewer 3 Report

This article is potentially interesting. For those who are not in the field, it is quite difficult to follow.

Authors names, affiliations, etc. should follow the template.

Line 28-29 have no meaning in the context. Consider removing and rephrase line30-33.

Line 60. Please give full term for ISDGMA and any other abbreviation when first used in the main text, as well as in the abstract.

A Materials and Methods section should replace Experimental Details, with subsections such as Synthesis, Resin preparation etc.

Figure 2 should be moved in the Introduction part in the text, where it is mentioned in the text.

Figures 6,7 and 8 are huge and in a landscape position. Please find a way to properly integrate them in the text, in a portrait position.

Please check the references style and modify it according to the template.

Additional information regarding Author Contributions, Funding, Conflicts of Interest, etc. should be added, according to the template.

Reviewer 4 Report

In this work, a series of isosorbide dimethacrylates with benzoate aromatic spacers was synthesized, photopolymerized, and investigated for potential application in dental restorative resins. The resulting polymers were evaluated for the degree of monomeric conversion, polymerization shrinkage, water sorption, glass transition temperature, and flexural strength.

The work done by the authors is interesting and valuable. However, I would like to address the following comments to the Authors:

  1. The Abstract states the bio-based nature of the synthesized isosorbide dimethacrylates. To prove this statement, the natural sources of the starting materials should be described in the Introduction part. What is the bio-based carbon content of the synthesized isosorbide dimethacrylates?
  2. City and country of supplier/manufacturer of the used chemicals and equipment are missing.
  3. What ratio of solvents was used in the eluent for column chromatography?
  4. The quality of Figures 6, 7, and 8 is insufficient.
  5. Does water sorption stop in all polymer samples after 7 days? It seems that the water sorption process does not stop at 7th day in Figure 15. The kinetic curves of water sorption vs. time showing the settled down values of all polymer samples are missing.
  6. The comparison of mechanical characteristics of all initial polymer samples and the maximally swollen polymer samples is needed. Immersion for 24 h (Figure 16) could be too short time for the maximal PBS (phosphate-buffered saline) sorption by the polymer samples.
  7. What is the final conclusion: is any isosorbide dimethacrylate investigated in this study suitable to replace bisGMA in dental resins?